# The Effects of Deregulated Ribosomal Biogenesis in Cancer

**DOI:** 10.3390/biom13111593

**Published:** 2023-10-30

**Authors:** Yiwei Lu, Shizhuo Wang, Yisheng Jiao

**Affiliations:** Department of Obstetrics and Gynecology, Shengjing Hospital of China Medical University, Shenyang 110055, China; 18840186962@163.com

**Keywords:** ribosomes, ribosomal biogenesis, tumorigenesis, metastasis, nucleolar stress regulation, cell signaling pathways

## Abstract

**Simple Summary:**

Ribosomes are macromolecular complexes responsible for mRNA translation. When ribosome biogenesis is hyperactive or aberrant or ribosomal factors are mutated, ribosomopathies may occur, elevating cancer risks. This review aims to discuss recent research regarding the complex mechanism responsible for regulating ribosome biogenesis and delineate how the deregulation of this process is connected to cancer pathogenesis. Providing our perspective on how these observations provide opportunities for designing new targeted cancer treatments. In doing so, we hope to draw attention to persisting gaps in the literature and candidate targets involved in ribosomal biogenesis for cancer therapies to facilitate further research in this field.

**Abstract:**

Ribosomes are macromolecular ribonucleoprotein complexes assembled from RNA and proteins. Functional ribosomes arise from the nucleolus, require ribosomal RNA processing and the coordinated assembly of ribosomal proteins (RPs), and are frequently hyperactivated to support the requirement for protein synthesis during the self-biosynthetic and metabolic activities of cancer cells. Studies have provided relevant information on targeted anticancer molecules involved in ribosome biogenesis (RiBi), as increased RiBi is characteristic of many types of cancer. The association between unlimited cell proliferation and alterations in specific steps of RiBi has been highlighted as a possible critical driver of tumorigenesis and metastasis. Thus, alterations in numerous regulators and actors involved in RiBi, particularly in cancer, significantly affect the rate and quality of protein synthesis and, ultimately, the transcriptome to generate the associated proteome. Alterations in RiBi in cancer cells activate nucleolar stress response-related pathways that play important roles in cancer-targeted interventions and immunotherapies. In this review, we focus on the association between alterations in RiBi and cancer. Emphasis is placed on RiBi deregulation and its secondary consequences, including changes in protein synthesis, loss of RPs, adaptive transcription and translation, nucleolar stress regulation, metabolic changes, and the impaired ribosome biogenesis checkpoint.

## 1. Introduction

Ribosomes are the intracellular machinery required for translating specific mRNAs [1]. This machinery synthesizes new proteins based on the sequence of mRNA codons, and precise correction is determined by correct folding by the ribosome [2]. Ribosomal components include ribosomal proteins (RPs) and ribosomal RNA (rRNA). Ribosome production, also known as ribosome biogenesis (RiBi), is initiated by the transcription of rRNA [3]. In eukaryotes, RiBi occurs in the nucleus and requires four rRNAs (5S, 5.8S, 18S, and 28S rRNA) and 80 RPs (79 in yeast) that coordinate their assembly [4]. rRNA transcription and pre-ribosomal subunit assembly occur in the nucleolus, and these subunits are subsequently exported to the cytoplasm, where translation of proteins takes place [5]. RPs play a crucial role in RiBi in addition to being ribosomal components [6]. RiBi is a complex multistep maturation process that assembles proteins and RNAs across multiple cellular compartments to produce mature ribosomes [7]. It is a multistep process of ribosomal DNA (rDNA)-restricted transcription in the nucleolus [8]. RiBi is highly coordinated in eukaryotic cells, and RNA polymerases (RNA Pol I, Pol II, and Pol III) are responsible for transcribing rDNA into rRNA [5]; RiBi transcribes 47S precursor rRNAs (47S pre-rRNAs) from rDNA genes via RNA Pol I in the nucleolus, 5S rRNA (nucleoli in human cells, nucleoplasm in yeast) is added as part of the 60S RNP complex to the 5S pregranule in the nucleoli, transcribed by RNA Pol III in the nucleoplasm and co-transcribed to 90S, RP mRNA is transcribed by RNA Pol II in the nucleus [9]. After nuclear output, assembly is made in the cytoplasm, including the release of remaining glycobiogenic factors (RBFs) and incorporation of missing RPs, resulting in mature, translationally capable subunits [10]. The rate-limiting and most crucial stage of RiBi is the transcription of 47S rRNA by RNA Pol I. SL1 contains TATA-binding protein correlated factors responsible for Pol I promoter specificity [11]. Selective factor 1 (SL1), the multi-HMGbox factor UBTF/UBF, and the RNA Pol I-related initiation factor RRN3 are the basic factors of the transcriptional process of mammalian RNA Pol I [8]. TATA-box binding protein (TBP) and RNA Pol I-specific TBP-related factors TAFIA to D make up the protein SL1. These TAFs’ RNA Pol I transcription function and species–specific function are both governed by their evolutionary diversity [12].

Deregulated signaling networks, metabolic reprogramming, and aberrant non-coding RNA expression can enhance RNA Pol I transcriptional activity in cancer cells [13]. The RNA Pol I transcriptional machinery is activated by oncogenes, such as MYC, and growth signaling mediators, such as the mechanistic target of rapamycin kinase (mTOR) [14]. RiBi is one of the most multifaceted and energy-consuming processes in biology [15], and as unlimited proliferation, invasion, and metastasis are the hallmarks of many cancers, cancer cells alter and boost their biochemical and metabolic activities, particularly RiBi, to meet their high energy requirements [16].

Many ribosome diseases are associated with increased susceptibility to cancer, and there is growing evidence that defects in RiBi can drive tumorigenesis [10]. A process known as the impaired ribosome biogenesis checkpoint (IRBC) can sense abnormal RiBi and subsequently control adaptive reactions and the destruction of faulty ribosomes [17]. However, the specific steps involved in RiBi regulation in humans remain poorly understood. Discovering their involvement in cancer etiology and creating cutting-edge anticancer medicines need a deeper knowledge of these processes. As such, in this review, we discuss recent research concerning RiBi regulation and provide an overview of what is known regarding deregulated RiBi and cancer.

## 2. Ribosomes in Cancer Pathogenesis

RiBi overoccurrence is a controllable target of cancer; rRNA accounts for about 80% of total RNA and RiBi requires a large number of nucleotides, mTORC1 or c-MYC oncogenic signaling pathway overactivation while upregulating RiBi and nucleotide de novo synthesis [18]. Excessive activation of RiBi can be triggered by the expression of oncogenes or loss of tumor suppressor genes [19]. Several oncogenic pathways, such as p53, phosphatidylinositol 3-kinase (PI3K)/AKT/mammalian target of rapamycin complex 1 (mTORC1), c-MYC, and epithelial cell transformation sequence 2 (ECT2), are stimulated at numerous levels during RiBi, further confirming the importance of RiBi hyperactivation in diseases [20]. Expression and disruption of proteostasis in ribosomopathies are linked to diseases caused by RP mutations or the dysregulation of factors associated with RNA Pol I-dependent transcription and rRNA processing, guaranteed by quality control checkpoints, bypassing misassembled, dysfunctional ribosomes that can result in reduced translational fidelity and thereby altering genes, resulting in the disruption of ribosome synthesis or assembly [21,22]. Some RPs have been identified as “cancer genes” in a variety of cancer types by large-scale sequencing research [23]. Understanding the importance of RP coding and ribosomal biogenic factor (RBF) gene mutations in human disease is critical. Perturbations in ribosome biogenesis, which is closely related to cellular activity, may modify p53 through MDM2 E3 ligase activity that is mediated by ribosomal protein (RP) [24]. Ribosomal RNA processing 15 homolog (RPL15) promotes metastatic growth in multiple cancer organs, and enhancing the altered expression of RPL15 may alter the translational efficiency of ribosomes [25]. RPL15 lacks inhibition of p53 degradation and also prevents MDM2 from attaching to p53. RPL15 mediates the progression of hepatocellular carcinoma (HCC) via the RPs–MDM2–p53 signaling pathway [25]. RPL15 is upregulated in a variety of cancer cells, such as gastric cancer, triple-negative breast cancer, and leukemia [10]. Multiple RPs interact with MDM2–p53, a tiny ribosomal protein that makes up 40SRNP. In the presence of ribosomal stress, RPS14 interacts with MDM2 and stabilizes p53 [26]. Among the targets associated with protein synthesis, there are mTOR and RPS6KA5 and several structural components of RPL15 (eL15), RPL35 (uL29), and RPL13 (eL13); these three structural ribosomal proteins (RP) aggregate on a solvent-exposed surface near the exit tunnel behind the 60S ribosome and interact physically [27]. There are differences in the translation and synthesis of mRNA to a protein under different pathological or physiological conditions, and the difference in the expression of a protein in the tumor may be due to the regulatory effect caused by its own mRNA in order to adapt to plasticity. The regulation of mRNA translation can partly explain the difference between protein abundance and its corresponding mRNA abundance; in fact, abnormal expression and activity of RBF, RP, RNA-binding protein (RBP) and non-coding RNA (ncRNA), as well as regulatory changes in RNA secondary structure, RNA methylation and the usage of codons or upstream open reading frame (uORF) selection that takes place there, highlights the role of translation control in the development of tumors [27]. The concept of plasticity (plasticity indicates both the ability to take forms and give forms) in mRNA translation has recently been applied to cancer, where it is regulated simultaneously by various mechanisms [28]. Through quick, targeted changes in the cellular proteome, this translational plasticity enables the start, progression, and resistance to anticancer therapy of particular cancer subtypes.

RRP15 is a critical nucleolar protein for RiBi and checkpoint control factor, and an RRP15 deficiency induces ribosomal stress through inhibition of the Wnt/beta-catenin pathway to suppress cell proliferation and metastasis, suggesting potential novel targets in patients with high-RiBi cancer [29]. β-catenin is recognized as an upstream activator of c-MYC expression, and β-catenin overexpression restores c-MYC expression as well as protein production in mouse C2C12 myotubes. For the first time, researchers discovered a direct relationship between β-catenin–c-MYC depletion and lower ribosome content and protein production [18]. Mechanistic targets of the Wnt/B-catenin, PI3K/Akt/mTOR, and Hedgehog (Hh) pathways, which inactivate the tumor protein p53, constitute a major risk factor for cancer development [30]. p53 may bind 4E-BP4 via the mTOR pathway or inhibit translation by titrating the amount of free eIF1E [31]. mTORC1 then promotes mRNA initiation and elongation through eIF4E and eIF2a [20]. MYC, mTOR, and extracellular signal-regulated kinase (ERK) stimulate RNA Pol I activity. These genes are frequently dysregulated during tumorigenesis, leading to increased RiBi. Loss of function of tumor suppressor proteins (e.g., p53, ADP ribose ribosylation factor (ARF), retinoblastoma protein (pRb), and phosphatase and tensin homolog (PTEN)) also leads to the dysregulation of RNA Pol I-dependent transcription [32]. However, how mutations in RPs or ribosomal biogenic factors promote cancer is still unknown [33].

The mTORC1/4E-BP/eukaryotic translation initiation factor 4E (eIF4E), a component of the eukaryotic initiation factor 4F complex, is a promising target for cancer treatment and prevention of drug resistance [34]. When nutrients in RiBi are restricted, or another off-signal is received, mTORC1, the main regulator of the mTOR pathway, is inactivated [35]. mTORC1 is a protein complex containing the kinase mTOR that phosphorylates 4E-BPs, reducing their affinity for eIF4E. Due to the lack of inhibitory 4E-BP binding, eIF4E and eIF4G interact to form the eIF4F complex [36]. One of the main regulators of RiBi also includes MYC, which, like ribosomes, induces transcription mediated by RNA Pol I, II, and III [37]. The oncoprotein MYC controls the activity of eIF4F by increasing the expression of all components of eukaryotic initiation factor 4F (eIF4F). In turn, MYC mRNA is a translation target for the eIF4F complex [38] (Figure 1). The effect of abnormalities in RiBi on cellular signaling pathways remains poorly understood. Understanding the significance of mutations in RP-coding and ribosomal biogenesis factor (RBF) genes in human diseases is crucial.

## 3. RP Gene Deletions Increase Susceptibility to Cancer

Altered expression of individual component RPs, such as RPL15, may alter the translation efficiency of ribosomes, either globally or against specific subsets of mRNA [39]. Human hematological and developmental problems, p53 activation, and cancer have all been linked to inactivating mutations in RP genes (RPGs) [40]. For example, knocking down specific RPGs in TP53 wild-type lung adenocarcinoma lines and leukemia cells resulted in the activation of p53 and four p53 target genes, P21 and BCL2-associated X-cell apoptosis regulator (BAX) [41]. Activation of the p53 pathway by reduced RPG expression most likely occurs via the 5S ribonucleoprotein (5SRNP)/murine double minute 2 (MDM2) pathway [42]. However, the release of many RPs from ribosomes into the nucleoplasm may occur in response to ribosomal stress induced by radiation, genotoxic agents, or oxidative chemicals, resulting in impaired RiBi and binding of these RPs to MDM2, thus inhibiting the degradation of p53 [43].

Genome-wide analysis of human cancers has shown that p53 mutations are tightly associated with the loss of RP expression or activity, particularly ribosomal protein L5 (RPL5) and L11 (RPL11) mutations, which have recently reinforced the link between ribosomal defects and cancer in multiple tumor types (Figure 2) [44]. Somatic mutations in RPGs are the possible drivers of several sporadic cancers. Other genes encoding RPs have somatic mutations in sporadic cancers, including ribosomal protein SA (RPSA; uS2), ribosomal protein S5 (RPS5; uS7), RPS20 (uS10), RPS27 (eS27), RPL11, RPL22 (eL22), and germline mutations in ribosomal protein L23A (RPL23A; uL23) [45]. In RP deficiency, RPs associated with the cell cycle/p53 signaling pathway are predominantly enriched in the nucleus and have a large impact on cancer cells. These findings may open new possibilities for cancer treatment in patients with TP53 mutations.

## 4. Translation and Transcription in RiBi and the Link to p53

The rate-limiting step in RiBi is the transcription of pre-rRNA genes by RNA Pol I [46]. The upstream transcription factor TATA-box binding protein (TBP) and four TATA box-associated proteins are components of the mammalian core RNA Pol I transcription factors upstream binding factor 1 (UBF1) and selective factor 1 (SL1) [47]. Polymerase-associated factor 53 (PAF53) and 49 (PAF49) are present in the active RNA Pol I subfraction during in vitro rDNA transcription, and PAF53 is rapidly degraded in response to a loss of PAF49 [48], which inhibits rDNA transcription. In disease models with deregulated RiBi and/or RNA Pol I-dependent transcription, PAF49 effectively inhibits rDNA transcription and p53 accumulation [49], causing a p53-dependent nucleolar stress response [50]. Mutations in the shared subunits of RNA Pol I and III or the RNA Pol I-associated factor treacle ribosomal biogenesis factor 1 (TCOF1) result in Treacher–Collins syndrome (TCS) [51]. Haploinsufficiency of TCOF1 perturbs mature RiBi, leading to the stabilization of cell cycle arrest mediated by p53 and cyclin G1 [52]. In tumor cells, TCOF1 is a nucleolar protein that is important for RiBi and promotes nucleolar translocation of glioma-associated oncogene 1 (GLI1) [53].

Human cancer is linked to the dysregulation of the Hh/GLI signaling pathway. Numerous cancers, including medulloblastoma, prostate cancer, basal cell carcinoma, colorectal cancer, and pancreatic cancer, activate this signaling pathway [49]. In mouse embryonic fibroblasts (MEF) harboring oncogenes, p53-dependent apoptosis and growth arrest are inhibited by Hh signaling [54]. Hh signaling is canonically activated through Hh ligand interactions with the transmembrane receptor patched 1 (PTCH1), which inhibits G-protein-coupled transmembrane receptor smoothened (SMO). Thus, the zinc finger transcription of GLI1 (the end effector of the Hh pathway) is released from the suppressor of fused (SUFU)-mediated suppression of cytoplasmic sequestration, allowing nuclear translocation and target gene activation [55]. Activation of the GLI family of transcription factors is induced by the dissociation of the cytoplasmic repressor protein SUFU, which leads to the transcriptional activation of GLI target genes [54]. Novel roles for nucleolar GLI1 in orchestrating the restoration of RNA Pol I activity have been identified [56]. Inhibition of Hh and RNA Pol I activities prevents lung cancer cells from growing. RiBi and Hh activity act as actionable signaling mechanisms. SMO activation after activation at baseline and after DNA damage inhibits p53 accumulation through Hh signaling, and MDM2 facilitates p53 ubiquitination through Hh signaling [53].

## 5. The 5S RNP–MDM2–p53 IRBC Pathway May Provide a Barrier to the Development of Cancer

Impaired ribosomal biogenesis checkpoints (IRBCs) and the RPL5/RPL11/5S rRNA–MDM2–p53 pathways are involved in tumorigenesis [57]. When rRNA synthesis, processing, and assembly regulate a change in RiBi rate, it triggers a stress response to p53 in nucleoli [58]. 5S rRNA, RPL5, and RPL11 form 5SRNP and are assembly intermediates of the large ribosomal subunit [59]. The altered RiBi causes an excess of 5SRNP and triggers IRBC [60]. Depletion of RiBi components also activates p53 via IRBC [60]. For example, HEAT repeat containing 1 (HEATR1) disrupts nucleolar architecture: HEATR1 deficiency leads to cell cycle arrest in a p53-dependent manner, and the p53-p21 checkpoint response of the RPL5/RPL11-MDM 2 axis activates nucleolar stress in the absence of DNA damage [61]. A change in RiBi promotes the recruitment of RPs (mostly RPL5/RPL11) and nucleolar factors, as well as binding to the central acidic domain of Mdm2, which interferes with its interaction with p53 [62]. In response to RiBi stress, RPL5 has been identified as a “cancer gene”, while RPL3 activates p53 without the assistance of p21 and causes cell cycle arrest and apoptosis [63].

Acute disruption of RiBi, which activates the nucleolar surveillance pathway (NSP), causes certain RPs to bind to MDM2 in an inhibitory manner, thereby increasing the amount of p53 in the nucleus [64]. The study found that 77.3% of 60S-specific RPs and 81.3% of 40S ribosomal subunits induced a “p53-positive” phenotype upon depletion, implying that RPs of either subunit had similar contributions to NSP p53 [33]. The well-known mechanism of genotoxic stress, which causes significant post-translational regulation of p53 and subsequently modulates its association with MDM2, is insufficient for stabilizing p53 in the absence of functioning NSPs [65]. Defects in the synthesis of ribosomal subunits 40S (SSU) and 60S (LSU) are responsible for the development of more than 20 hereditary illnesses (ribosomopathies) and numerous malignancies [64]. 5S RNP, as an intermediary in the assembly process of LSU [66], becomes a key regulator of NSP through RPL5 (uL18) and RPL11 (uL5). Following recruitment of the nucleolar components ribosome production factor 2 homolog (Rpf2) and ribosome biogenesis regulator 1 homolog (Rrs1), the developing 5S rRNA attaches to the nuclear import complex symportin 1 (Syo1)-uL18-uL5 and develops into 5SRNP precursors that can assemble into pre-ribosomes [33,67]. Syo1 is a component of the emerging 5S RNP [68]. However, in the circumstances (such as nucleolar stress) that increase the free 5S RNP pool, Syo1 and Rpf2-Rrs1 may not be present in sufficient amounts or may be prevented from effectively competing for Mdm2 binding [67]. 5S RNP decreased mature SSU levels due to defective SSU production and p53 activation by stimulating RP synthesis, leading to 5S RNP overproduction. This means that changes in mature ribosome levels are not necessary for p53 induction for issues with SSU production and that faults in the generation pathway itself may result in 5S RNP-mediated p53 activation [66]. It is hypothesized that either a factor or multiple components involved in RiBi connect these two pathways, though further research is required. Both routes are necessary for the creation of this early nucleolar component, which may be rate-limiting, but if SSU production is stopped, it is trapped in the SSU processome complex and is not accessible for LSU synthesis.

By blocking MDM2 mRNA production, triptolide (TP) treatment inhibits MDM2 at the transcriptional level [32]. During ribosome stress, TP treatment results in translocation and nucleolar disintegration of RNA Pol I and upstream binding factor (UBF) [43]. This and other evidence suggest that the 5S RNP–MDM2–p53 IRBC pathway can be used as a measure to prevent cancer growth [69]. However, the mechanism of action remains unclear and requires more thorough study. The two unresolved issues are the nature of the damage that induces IRBC and the signal that mediates this reaction. The IRBC reaction after the occurrence of damage to ribosome organisms is attributed to nucleolar destruction, and the depletion of RPS6 eliminates the production of the 40S subunit, causing the induction of p53 but having no impact on the nucleolar architecture or the synthesis of the 60S subunit. Future research will focus on clarifying the function of the 60S ribosomal subunit structure in mediating IRBC. A highly potent p53 activation signal is blocking RiBi. BMH-21 and CX-5461 are two RNA Pol I inhibitors that have been investigated [70]. These substances, other than nucleolar rDNA, may interfere with other DNA-related activities, like DNA intercalators (BMH-21) or TOP2 inhibitors (CX-5461).

## 6. mTOR Regulation of RiBi

mTOR controls RiBi via rRNA and RP synthesis through transcription and translation, two major mechanisms (Figure 3) [71]. In vertebrates, RP production is primarily controlled by the translation of RP mRNA [72]. mTOR can boost translation by suppressing the activity of the RP S6 kinases (S6K1 and S6K2) and phosphorylation of the eIF4E-binding proteins (4E-BP1 and 4E-BP2) [73]. Through the mTOR-dependent translation of 4E-BPs, phosphorylation of 4E-BP selectively upregulates the translation of terminal oligopyrimidine (TOP) motif-containing mRNAs and encourages RiBi and subsequent protein synthesis [74]. The 5′ TOP motif is an essential characteristic of RP mRNA sequences in various vertebrates that mediates RiBi [59]. 5′TOP motifs in endowing mTOR-dependent translation regulation [73]. The 5′ end of mRNAs that are bound by eukaryotic translation initiation factor (eIF) complexes (eIF4E, eIF4G, and eIF4A) during translation initiation contain the 5′ TOP motif [75].

Ribosomal protein S6 (RPS6) mediates the interaction of 5′ TOP transcripts during translation initiation. Decreased phosphorylation of RPS6 may affect overall mRNA translation initiation efficiency, but containing a 5′ TOP motif could avoid this effect [76,77]. eIF4E affinity is therefore reduced, as demonstrated by reduced 5′ TOP mRNA translation in 4EBP1/2 double knockouts [78]. Both S6K and 4EBP1/2 are considered 5′ TOP motif-specific factors [76], and phosphorylation of S6K can activate multiple effectors. For example, the pS6K-mediated phosphorylation of UBF and tripartite motif containing 24 (TIF1A) upregulates the transcription of rRNA genes through RNA Pol I [79].

### 6.1. mTORC1-Regulated RiBi Is Involved with Tissue Regeneration and Cancer Development

It is crucial to control the translation and stability of 5′ TOP mRNA throughout both healthy cell growth and disease progression; however, it has been challenging to understand the molecular processes that underpin the regulation of these processes. Any inhibitor affecting the fundamental functions of RiBi is anticipated to have a global pleiotropic influence [80]. mTORC1-dependent translational reprogramming is critical for mTORC1-driven cellular processes [81]; more than 90% of mTORC1-targeted mRNAs include the 5′ TOP more or the pyrimidine-rich translation element (PRTE), suggesting that the 5′ TOP motif or PRTE is a predictor of mTORC1 reliance [82]. The relationship between mTOR-dependent translation reprogramming and cancer in RiBi still needs further study.

As previously stated, RP production in higher eukaryotes is greatly regulated at the translational level, and this regulation is controlled by the 5′ TOP motif, which is situated at the transcriptional start site of RPGs [82]. Another explanation may be that cellular translation upregulates mRNAs to which RPL5 and RPL11, both 5′ TOP motif-containing factors, are bound and present in quantities that are higher than those necessary for 60S RiBi, causing the ribosome-free RPL5/RPL11/5S rRNA complex to form, which then causes p53 to become active [83]. Any interference with RiBi may decrease the pre-ribosome’s ability to interact with the developing RPL5/RPL11/5S rRNA complex, causing it to accumulate in a free form [62]. Deletion of 40S subunit RPs can abolish the RPL5/RPL11/5s rRNA complex, which is not included in the assembly of homologous ribosomal subunits. However, the buildup of the 60S portion of 47S rRNA is what triggers the formation of the ribosome-free RPL5/RPL11/5S rRNA complex and p53 activation. However, this process may be slowed by the loss of 40S RPs assembled into the 90S pre-ribosome [84].

La-associated protein 1 (LARP1) has been identified as a possible major specificity factor controlling the translation of 5′ TOP mRNA by preventing the eIF4F complex from forming on these transcripts. Thus, through LARP1, mTORC1 regulates the translation of TOP mRNA [85,86]. mTORC1 phosphorylates LARP1 both in vitro and in vivo to coordinate protein synthesis via mTOR signaling; rapamycin and Torin 1 efficiently block these effects. LARP1 and poly (A) binding protein cytoplasmic 1 (PABPC1) are more tightly bound when mTORC1 is inhibited, resulting in translational repression of TOP mRNA binding and translation [86].

Significant enrichment of 5′ TOP tracts is found in preferentially translated genes [87]. We discovered that highly translated squamous cell carcinoma transcripts use 5′ TOP and PRTE motifs more frequently, with typically short 5′ UTRs and less exposure of RNA secondary structures, on a genome-wide scale [86]. For ribosome recruitment and translation to begin, the 5′ UTR is necessary [88]. mRNAs with a long and highly structured 5′ UTR are more susceptible to ribosome depletion [73]. The identical proteins encoded by the two RP transcripts RPL21 and Rpl29 have the same 3′ UTR and differ only with respect to their 5′ UTRs, which differ further in individuals with human head and neck squamous cell carcinoma (HNSCC); in fact, Rpl21 and Rpl29 show increased overall translational efficiency in squamous cell carcinoma cells [86,89]. As such, certain 5′ UTR isoforms may be associated with disease progression in cancer patients [86]. A small group of genes with 5′ UTR isoforms showed improved overall translational efficiency, setting their responsiveness to food-sensing mechanisms that are mTORC1-dependent and directing the translational potential of mRNAs to alter protein synthesis rates through precise 5′ UTR sequences [73]. Specific carcinogenic transcripts with 5′ UTR secondary structures are translated more quickly thanks to EIF4A in T-cell acute lymphoblastic leukemia [90]. Increased eIF4E levels promote malignant transformation by being translated into specific oncogenes (e.g., the gene encoding cyclin D1) [91]. Since they govern the translation of certain mRNAs downstream of oncogenic signaling cascades, eIFs, which control the initiation phase of translation, are crucial for tumor growth [92]. Genes involved in alternative splicing in squamous cell carcinoma are rich in RPs and splicing factors, including ribosomal protein L38 (RPL38). Twelve of the 23 RPs with variable splicing have alternative 5′ UTR isoforms [89]. The identified alternative isoforms are mainly composed of 5′ and 3′ UTR alternative splice sites, which overall increase their rate of protein synthesis. [86]. The ability of the 5′ UTR of TOP transcripts to promote preferential expression in the presence of nonstructural protein 1 (Nsp1) has been confirmed [87].

### 6.2. mTORC1-Directed RiBi Influences Cancer Stem Cell Function

Stem cell upregulation by RiBi [93] suggests that the mTOR pathway is overactivated in cancer stem cells (CSC) [94]. How can stem cells maintain modest protein synthesis rates in the context of mTOR signaling? Regulation of mTOR signaling is essential for the coordination of protein synthesis, RiBi, and stem cell activity; however, high RiBi is not immediately evident. Stem cells have a shortened 3′ UTR compared to differentiated cells, and 3′ UTR shortening is a common cause of the loss of translational control over carcinogenic mRNAs in cancer [95]. It has been demonstrated that intestinal stem cell senescence is driven by mTORC1 through the p38 mitogen-activated protein kinase (MAPK)/p53 pathway [96].

mTORC1 plays a direct role in cancer stem cells through subsequent impactors that function in RiBi and protein synthesis at various stages. Research indicates that the impact of mTOR inhibitors on CSC may rely on the genetic background and rewiring of cancer stemness pathways; however, the connection between mTOR inhibitors and CSC is complicated, and their further relationship to RiBi [97] or tumor cell resistance [98] has not yet been reported.

### 6.3. MYC-Induced Impaired IRBC May Be a Potential Target for Cancer Therapy

#### 6.3.1. Antagonizing RiBi Effects of c-MYC in Cancer

MYC is a modulator of RiBi and one of the proto-oncogenes involved in aberrant RiBi [91]. Oncogenic MYC-driven tumors induce RiBi hyperactivation [99]. The c-MYC transcription factor elevates RiBi by regulating RNA pol II-dependent transcription [100], and its overexpression leads to cell growth and hypertrophy with increased RiBi [101]. The creation of eIF4E is stimulated by the overexpression of c-MYC, which facilitates the translation of structured mRNA into oncogenes that promote cell proliferation and provide tamoxifen resistance [102].

MYC stimulation results in steady p53 protein augmentation and ribosome biogenesis. c-MYC has been shown to inhibit TP53 by c-MYC-induced long noncoding RNA inactivation p53 (MILIP) [103]. Unexpectedly, IRBC complexes bound to HDM2 increased significantly, although free RPL5/RPL11 levels remained unchanged. Mdm2 gene amplification is the main mechanism of oncogenic activation [104]. MYC induction enhances the RiBi and stability of the p53 protein, while MYC silencing reduces the expression of RPL5, RPL11, and 5S rRNA in IRBC complexes [105], resulting in a rapid decrease in the p53 protein half-life in a human double minute 2 (HDM2)-dependent manner [106]. RPL5 and RPL11 inhibit MYC transcription and destroy the stability of MYC mRNA, forming a negative feedback loop [104,107]. IRBC may be an important anticancer barrier. It is unclear whether this reaction is mediated by IRBC and what molecular process promotes increased association between RPL5/RPL11/5S rRNA complexes and HDM2 after MYC overexpression [107]. It is critical to understand how this signaling pathway engages all p53 transcriptional targets and consider how they affect the body’s defense mechanisms against various diseases by influencing cellular adaptive responses to RiBi stress, mainly cancer. This pathway could serve as a defense against MYC-driven malignancies without p53 [107].

#### 6.3.2. MYC Drives Metabolic Redistribution in Cancer

MYC is a significant regulator of glycolysis, glutamine metabolism, nucleotide biosynthesis, and other metabolic activities. It is one of the most frequently dysregulated oncogenes in cancer [108]. Proto-oncogenes (including MYC) and certain pathogenic stressors (e.g., hypoxia) directly drive this process, which is necessary for malignant transformation or cancer cell survival [109]. Wnt- and TGF-induced ribosome RNA transcription, as well as the expression of genes involved in ribosomal subunit composition and nuclear–cytoplasmic output, are induced in response to the expression level of c-MYC [110]. The expression of metabolic enzymes (BCAT1, PYCR1, ASNS), transcription factors, and amino acid transporters (SLC1A5, SLC7A5, SLC7A11) was reduced by CM272. RPS6 is a gene involved in ribosome synthesis [111,112]. CM272 also significantly inhibits colony generation and migration of Hepatoblastoma (HB) cells. c-MYC plays a central role in HB, and CM272 not only downregulates c-MYC mRNA levels but also decreases the half-life of the c-MYC protein [111]. Cancer cells alter their metabolism to promote their growth. Bond reactions include the stimulation of amino acid absorption and synthesis, non-essential amino acids, RiBi, protein synthesis, and autophagy rescue, as well as additional events that assist macromolecule synthesis [112,113]. Small-cell lung cancer (SCLC) caused by MYC preferentially stimulates the mTOR and polyamine synthesis pathways, both of which are controlled by arginine [114].

Oncogenic kinases, in collaboration with MYC, can cause cancer cells to undergo metabolic reprogramming to actively sustain the increasing demand for the resources needed to increase cell mass and improve DNA replication and cell division. MYC promotes guanosine triphosphate (GTP) production and increases susceptibility to GTP inhibitors, such as those that target synthase inosine monophosphate dehydrogenase (IMPDH) (IMPDH is an enzyme in guanosine triphosphate (GTP) biosynthesis) [115,116]. Inosine 5′-monophosphate dehydrogenase (IMPDH) is a rate-limiting enzyme that catalyzes the dependent oxidation of inosine monophosphate (IMP) to xanthine monophosphate (XMP) by nicotinamide adenine dinucleotide (NAD), which is an important step in de novo biosynthesis of guanine nucleotides [117]. Purines activated by MYC biosynthesis genes such as IMPDH2 are co-expressed with ribosomal genes and required for maximal protein synthesis and growth rate in oncogenic MYC cells. We showed that RNA Pol I requires IMPDH-dependent GTP synthesis to bind to rDNA and produce pre-rRNA, joins two gene expression programs of MYC (RiBi and nucleotide biosynthesis), and inhibits when the supply of nucleotides and RiBi is insufficient. GTP works via GPN-loop GTPase 1 (GPN1) and -3 (GPN3), and MYC-dependent production of these two GTPases is essential for increasing RiBi. GTP, the final product of the IMPDH pathway, serves as a gate for RNA Pol I localization to rDNA and pre-rRNA synthesis [118]. This nucleotide–protein coordination is similar to MYC’s actions on phosphoribosyl pyrophosphate synthase-2 (PRPS2), another nucleotide biosynthesis enzyme. By binding with TFIIB and RNA Pol-III, MYC can also activate the transcription of 5S rRNA [119].

## 7. Beta-Catenin and RiBi

RRP15 deficiency-induced ribosomal stress promotes TOP mRNA LZTS2 (leucine zipper tumor suppressor) translation and leads to the nuclear export and degradation of beta-catenin, thus inhibiting Wnt/beta-catenin signaling in rectal cancer [120]. RRP15 knockdown inhibits Wnt/beta-catenin signaling in cells with or without p53, whereas p53 deletion only partially reverses inhibition of RRP15-directed Wnt/beta-catenin targeted transcription [121]. p53 stimulates WNT3 transcription, which activates the WNT/beta-catenin pathway in Adenomatous colorectal polyposis (APC) [122]. As such, RRP15 may be a potential new target for patients with high-RiBi CRC [123].

The most important ribosomal protein in carcinogenesis, Rpl15, increases the risk of tumor development in offspring derived from IVF (in vitro fertilization), and the way that overactive ribosome organisms in IVF-derived embryos react to reactive oxygen species (ROS) is linked to a number of traits. Nucleolin’s expression is increased by ROS, n-MYC, Ddx17, and Syne2. Oxidative damage establishes the oncogenetic cooperation between β-catenin, ribosomes, and TGF-β1 signaling pathways [124]. A decrease in ribosomal proteins is accompanied by a decrease in the expression and activity of β-catenin [125].

T antigen transcription in lens tumor cells targets viral carcinogenesis, cancer microRNAs, focal adhesion, p53, vascular endothelial growth factor (VEGF), metabolism of fructose and mannose, RiBi, choline and pyrimidine, and the PI3K–Akt and box O signaling pathways [126,127]. T antigen has a positive effect on the Wnt/beta-catenin pathway, and physical interactions between the central domain of T antigen and the C terminus of beta-catenin enhance beta-catenin levels and promote its nuclear import [127,128]. T antigen engages Rac family small GTPase 1 (Rac1) to prevent ubiquitin-dependent proteasomal degradation of beta-catenin. T antigen and beta-catenin nuclear co-localization dramatically increase T-cytokine-dependent promoter activity, as well as c-MYC and cyclin D1 activation [129]. Ribosomal protein L19 (RPL19) is an RP of the L19 family and a component of the 60S subunit. T antigen binding to RPL13, RPL15, RPL19, RPL29, and RPL25 was confirmed by co-immunoprecipitation using RPL19 and T antigen antibodies, indicating that T antigen promotes RiBi through the Wnt/beta-catenin pathway [127].

Lower expression of c-MYC, a transcriptional activator of c-MYC, and beta-catenin, a positive regulator of ribosome synthesis, are both linked to reduced ribosomal content. GSK3 removal does not lessen ammonia-induced beta-catenin breakdown, in contrast to how GSK3 normally regulates beta-catenin degradation. Hyperammonaemia does not alter beta-catenin expression or phosphorylation.

## 8. RiBi Selective Inhibitor

The quadruple structure of G-quadruplex (G4) DNA, which is produced by a series of guanine-rich DNA sequences, is thought to be a unique and potential target for a variety of cancer treatment formulations [130]. Two G4 ligands, CX-3543 and CX-5461, both produced from fluoroquinolones, have begun human clinical studies [131].

Quaflosin, another name for CX-3543, specifically breaks nucleolin/G4 complexes in the nucleus, limiting RNA polymerase I transcription and causing cancer cells to undergo apoptosis [132]. The first G-quadruplex-targeted medication clinical studies for CX-3543 for carcinoid/neuroendocrine tumors have gone to phase II. It targets small molecules that limit rDNA transcription [133], and clinical studies have employed it (NCT00955786, NCT00780663, NCT00780663). Tamoxifen resistance is conferred through a variety of routes by the over-surface of c-MYC, which also promotes the growth of ER+ve breast cancer. Increasing the biogenesis of ribosomes and producing mature ribose are important mechanisms [102]. RNA polymerase I (Pol I) small molecule targets are specifically attacked by CX-5461 [68]. There are still open clinical studies of CX-5461 for solid tumors (NCT02719977 and NCT04890613). RDNA transcription was reduced in patients who underwent CX-5461 treatment [134]. Cancers that can be cured include melanoma, breast cancer, acute myeloid leukemia, prostate cancer, and ovarian cancer [135,136,137]. CX5461 prevents nucleolar proteins or selective factor 1 (SL-1) from interacting with rDNA promoters, hence preventing rDNA transcription. [138] Cancer cells with BRCA mutations and CX-5461 are very deadly to polyclonal patient-derived xenograft models, including cancers resistant to PARP inhibition. A sophisticated Phase I clinical study using CX-5461 is now enrolling patients with malignancies caused by BRCA1/2 deficiency (Canadian Trials, NCT02719977, 2016) [137].

CX-5461 has been shown to successfully induce the expression of thyroid differentiation markers like PAX8, THRA, THRB, and NIS in human and murine thyroid tumors. It has also been shown to restore iodine uptake in thyroid tumors [139]. The G4 region in the c-MYC promoter contributes 80–90% of c-MYC transcription and has a 46 bp G4 sequence (known as Pu46) upstream of the c-MYC P1 promoter, CX-5461 functions in BRCA1/2, PALB2, or homologous recombinant defective mutant solid tumors [140].

The administration of copper-CX5461 in the form of DMPC/Choleliposomal formulation (DMPC/Chol Cu-CX2) enhanced efficacy and lengthened survival in BRCA-normal (BxPC3) and BRCA-deficient (Capan-1) pancreatic cancer mice [141]. In mice harboring cancer, the mTORC1 inhibitor everolimus was successfully used with CX-5461 to provide synergistic antitumor effects [134]. The pan-PIM kinase inhibitor CX-6258 and the abiraterone and enzalutamide-resistant PDTX models of prostate cancer exhibit significant tumor suppressive effects [142]. Given that it has been demonstrated to be effective against TP53 wild-type and mutant cancers, does CX-5461 promote p53 stability? Since nucleoli stress signaling in hematologic malignancies is highly dependent on p53 function, more studies should be conducted in this area. In two human xenograft osteosarcoma (OS) models and two allogeneic transplantation models for mice, the growth of TP53 wild-type and mutant tumors was assessed, and both types of OS tumors were suppressed by CX-5461 [31].

High-throughput cell screening was performed by BMH-21 to discover a brand-new p53-activating substance [29]. As a result of interactions between BMH-21 and BA-41 and the DNA G-quadruplex structure, as well as the c-MYC G-quadruplex, the expression of c-MYC is downregulated in human tumor cells [143]. CX-5461 prevents the beginning of Pol I transcription, whereas BMH-21 kills the largest portion of Pol I by intercalating DNA into GC-rich regions of rDNA and then generating signals. The transcriptional elongation step is also significantly impacted by BMH-21 processing, which in turn decreases Pol I occupancy on rDNA and transcriptional elongation in live cells [143]. In tumor cell lines, ex vivo prostate tissue cultures, and animal models, BMH-21 has antiproliferative properties [29]. The mechanism through which BMH-21 suppresses rRNA synthesis is unknown [144]. BMH-21 has the same effects on Pol I transcription and A190/RPA194 degradation in both yeast and humans. Investigations into the yeast system showed that the BMH-21 treatment’s most vulnerable stage is Pol I transcriptional elongation [145]. RPA194 stability is impacted by the Pol I transcription inhibitor BMH-21, which also produces elongation stress and is affected by RPA194 basic expression but minimal drug-induced protein turnover by RPA12. The Pol I transcription cycle’s beginning, extension, and end are all dependent on RPA12 [146].

What is BMH-21 specific for Pol I?

BMH-21 suppression could only affect Pol I transcription in chromatin tissues that contain rDNA. Nucleosomes are missing from our in vitro transcription DNA templates; therefore, it could be comparable to the rDNA that Pol I transcription in vivo. Unlike chromatin-bound genes that Pol II transcribes [147], ordered nucleosomes are absent in active rDNA repeats transcribed by Pol I [144]. Because Pol I has a faster rate of nucleotide addition and a less stable elongation complex than Pol II, Pol I may be more susceptible to the BMH-21 intercalation site. In contrast to BMH-21, CX-5461, which does not harm DNA, may damage DNA by stabilizing G-quadruplex DNA [137]. When topoisomerase II (Top2) is poisoned, CX-5461 produces cytotoxic effects instead of causing cell death by interfering with Pol I activity, according to an extensive, multidimensional analysis of the drug’s action [148].

We discovered that BMH-21 reduced the transcriptional elongation of Pol I and Pol III. The little influence on Pol III was a significant discovery even though Pol I was the most severely restricted [149]. Like Pol I, Pol III transcription is closely related to translation level [150]. Large ribosomal subunits are formed via the synthesis of 5 S rRNA by Pol III, and Pol III transcription is dysregulated in cancer cells [151,152].

BMH-21 inhibits the SKOV3 ovarian cancer cells’ capacity to survive in a dose-dependent manner [151]. The nuclear output and creation of the nucleolar stress markers nucleolar protein, ribophosphorus, and fibrillar protein are stimulated by BMH-21. Additionally, the tumor suppressors p53 and p15 were expressed at higher levels and their levels of protein were phosphorylated at serine 53 (p-p53-Ser15) by BMH-21 [153,154]. These findings are the first to show that BMH-21-induced p53-dependent nucleolar stress causes apoptosis in ovarian cancer cells [154].

The HaloTag selective labeling approach was used to uncover compounds that reduced the number of newly produced ribosomes in A375 malignant melanoma cells in addition to two high-throughput screens. Ribosome biogenesis inhibitors 1 and 2 (RBI1 and RBI2) are potent inhibitors of rRNA synthesis equivalent to the previously published ribosomal biogenesis inhibitor CX-5461 [155]. Lowering pre-rRNA levels inhibits RBI2, which is a mediator of ribosomal biogenesis. Although RBI2 treatment seems to have little effect on the transcriptional initiation or extension phases of Pol I, it does seem to boost the polyadenylation of rRNA [156].

## 9. Future Perspectives Regarding Activating Nucleolar Stress in Cancer Therapy

In the early stage, the mechanism of action of p53 in ribosomal biogenesis and cancer has been sufficient; for more molecular mechanism research and the role of cancer, specific target pathways, and changed or missing ribosomal protein research is still very lacking. Ribosomal-related changes, obtained knowledge about the role of dysregulated RiBi in human cancer and the mechanism of drug resistance to RiBi inhibitors are essential in cancer. Exome sequencing has revealed that a variety of malignancies, including leukemia, are linked to mutations in RPL5 (uL18), ribosomal protein L10 (RPL10; uL16), RPL11 (uL5), ribosomal protein L22 (RPL22; eL22), ribosomal protein S15 (RPS15; uS19), and ribosomal protein S20 (RPS20; uS10) [144]. In addition, the RiBi factor NPM1 (B23) is mutated in approximately one-third of all leukemia cases [157]. Oncogenic signaling may trigger different post-translational events on RP and rRNA and can also modulate the ribose interactome to shift ribosome properties into cancer-promoting modalities and the development of more realistic RiBi inhibitor combinations to increase effectiveness and/or eliminate cancer cell drug resistance will promote the development of more plausible RiBi inhibitor combos. The creation of predictive biomarkers will guide such future medicines in order to maximize the benefits for cancer patients.

## Figures and Tables

**Figure 1 biomolecules-13-01593-f001:**
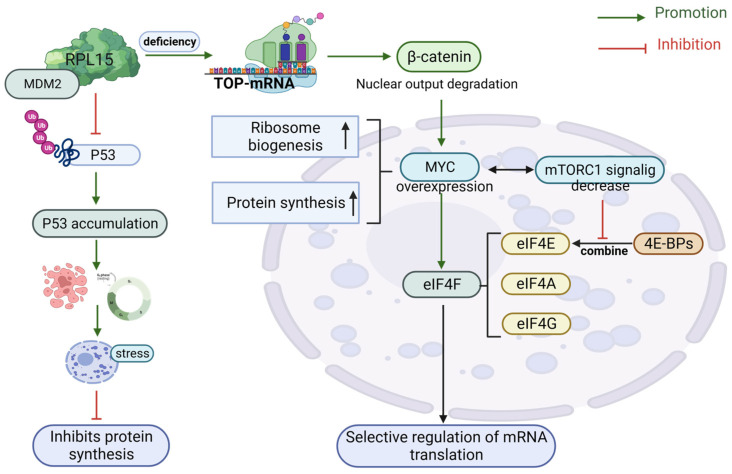
Ribosome protein modulates RiBi’s controllable target.

**Figure 2 biomolecules-13-01593-f002:**
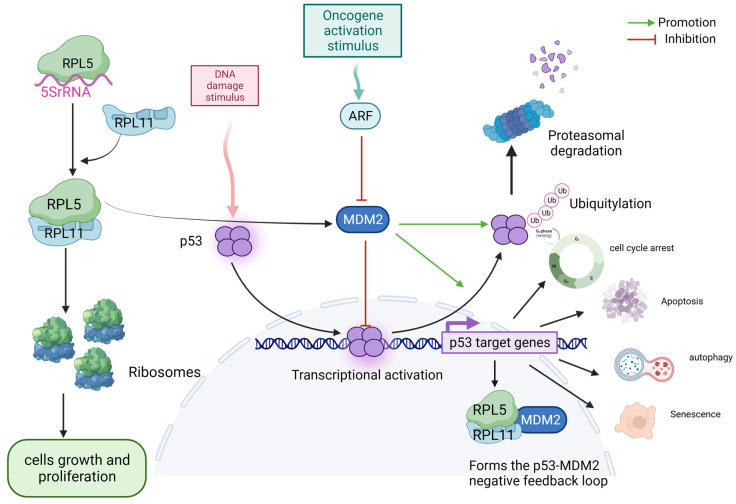
Stimulation of and involvement with the p53 pathway by ribosomal proteins. Abbreviations: ARF, ADP ribose ribosylation factor; MDM2, murine double minute 2; RPL5, ribosomal protein L5; RPL11, ribosomal protein L11.

**Figure 3 biomolecules-13-01593-f003:**
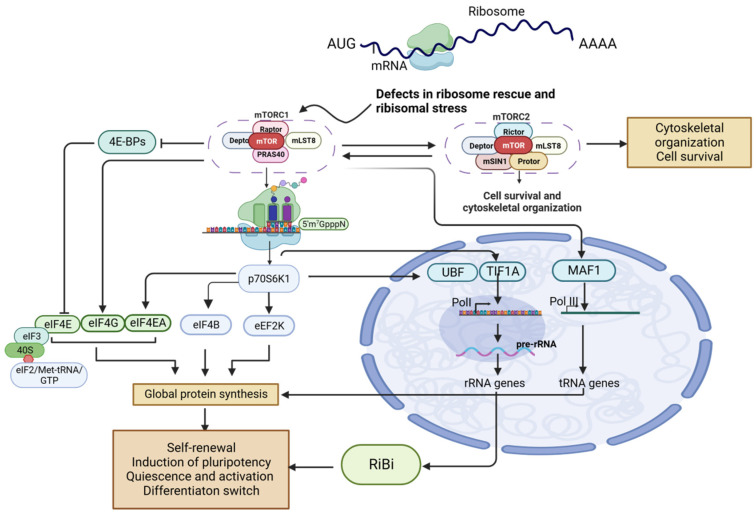
Factors involved in mTOR-directed regulation of ribosomal biogenesis. Abbreviations: elF-, elongation factor; GTP, guanosine triphosphate; mTOR, mammalian target of rapamycin; Pol, (RNA) polymerase.

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
