# Peer review of "The Effects of Deregulated Ribosomal Biogenesis in Cancer"

_biomolecules, 2023, doi:10.3390/biom13111593_

Round 1
Reviewer 1 Report
This is a very comprehensive review of the events accompanying ribosome biogenesis, of the involved factors and of the defects that may affect the process. Also, the authors cover extensively the literature about ribosomopathies and their link with cancerogenesis. However, the article is poorly readable, lacks a definite focus and contains numerous repetitions that make reading difficult. I think the authors should make an effort to shorten the text and make it more concise, highlighting the points they consider more important and on which future research should concentrate.
The quality of language is generally acceptable, although a through revision by someone proficient in English would greatly help in improving the manuscript. Special care should be given to the last paragraph which is almost incomprehensible.
Author Response
October 4th, 2023
Professor
Editor-in-Chief
Biomolecules
Dear Editor.
This article is poorly readable, lacks a clear focus, and contains a great deal of repetition that makes reading difficult.
Content that is repetitive and poorly relevant to the topic of this article has been removed, and the entire text has been substantially revised, with the changes highlighted in the body of the article
highlighting points that they felt were more important and where future research should focus.
This paper focuses on the involvement of P53 mTOR C-MYC β-linker proteins
in the regulation of RiBI and changes in cancer, and to add the status and progress of RiBi inhibitors. In the author's opinion, future research should focus on the development of RiBi inhibitors in these targets and regulatory mechanisms, as well as the improvement of clinical trials.
Special attention should be paid to the last paragraph, which is almost incomprehensible.
The content of the last paragraph has been revised in the hope that it will be better understood
Yiwei Lu

Reviewer 2 Report
This is a good review on the subject. However, there is a significant flaw that needs to be remedied before this review should be published. Despite citing 165 papers and specifically stating in Simple Summary: "This review aims to discuss recent research regarding the complex mechanism responsible for regulating ribosome biogenesis and delineate how deregulation of this process is connected to cancer pathogenesis. Providing our perspective on how these observations provide opportunities for designing new targeted cancer treatments.” the review does not address current progress on discovery and development of selective inhibitors of RiBi, such as CX-3543, CX-5461 and BMH-21. This needs to be fixed. There are some other minor things:
1. Typo in Figure 2. Eukaryotic LSU contains 28S, not 18S rRNA. It also contains both 5.8S and 5S rRNAs
2. Lines 51 and 52: “47S pre-rRNA and 5S rRNA are transcribed in the nucleoplasm by RNA Pol III” - this is wrong, only 5S is
3. Line 55 “45S” should be “47S”
4. Line 147 – reference 156 is out of order
5. Line 171 – reference 157 is out of order
6. Line 183 – missing period
Author Response
October 4th, 2023
Prof. Dr.
Editor-in-Chief
Biomolecules
Dear Editor
- For your mistakes: “1. Typo in Figure 2. Eukaryotic LSU contains 28S, not 18S rRNA. It also contains both 5.8S and 5S rRNAs.
The revised picture is as follows, but because other reviewers think that the content here is redundant, it has been deleted in the original article, thank you again for correcting
The revised picture is as follows, but because other reviewers think that the content here is redundant, it has been deleted in the original article, thank you again for correcting
- Lines 51 and 52: “47S pre-rRNA and 5S rRNA are transcribed in the nucleoplasm by RNA Pol III” - this is wrong, only 5S is
- Line 55 “45S” should be “47S”
- Line 147 – reference 156 is out of order
- Line 171 – reference 157 is out of order
- Line 183 – missing period”
Corrections have been made in the original article, but as other editors have suggested deleting the content and images here, they have been removed in the original article and the changes are shown above. Booking highlighted, thank you very much for your careful attention!
This review does not address current progress in the discovery and development of RiBi selective inhibitors such as CX-3543, CX-5461 and BMH-21.
The purpose of this section is to describe the nature and research status of several inhibitors and their applications in cancer. We look forward to your corrections if there are any errors. On pages 11-12 of the manuscript

Reviewer 3 Report
In this work by Yiwei Lu et al entitled “The effects of deregulated ribosomal biogenesis in cancer”, the authors aim to review some of the factors and pathways that may be highjacked during carcinogenesis and are involved in ribosome biogenesis. Their work is focused mainly around four pillars P53, mTOR, MYC and Beta-catenin. The information provided throughout the review is sound and offers some interesting links between different ribosome proteins, their expressions and how the four different factors may affect their roles in cancer setting.
The biggest drawback of this work is the reasons why these four pillars (P53, mTOR, MYC and Beta-catenin) were chosen. No real case is presented to indicate why these factors should be prioritised over others. Nothing really links them all and one has to wonder why they should be focused on rather than others. It would be important for the authors to justify their choices and explain the rational here rather than offering a random collection of sections. This is even more important when reviewing the information offered in relation to MYC or even Beta-catenin, as the links there between their changes and consequential/coincidental changes in ribosome biogenesis are very scares and significantly stretched at best.
Other points to consider
There is no need to provide information about prokaryotes ribosome (Figure 2 and throughout) as it is really out of context in the big scheme of this work.
Some of the early text offer relatively basic information about the ribosome and it components. This background information has been common knowledge for well over 50 years and could be summarised further or just referenced by using some of the seminal reviews and books related to this topic.
Headers and footers suggest the manuscript is being considered in Cancers and not Biomolecules.
Author Response
October 4th, 2023
Prof. Dr.
Editor-in-Chief
Biomolecules
Dear Editor
- The biggest drawback of this work is the reasons why these four pillars (P53, mTOR, MYC and Beta-catenin) were chosen. No real case is presented to indicate why these factors should be prioritised over others.
The details have been modified in” 1. Introduction 2. Ribosomes in cancer pathogenesis and highlighted. The following picture has been added for better illustration.
- There is no need to provide information about prokaryotes ribosome (Figure 2 and throughout) as it is really out of context in the big scheme of this work.
The picture and part of the content have been removed
- Some of the early text offer relatively basic information about the ribosome and it components. This background information has been common knowledge for well over 50 years and could be summarised further or just referenced by using some of the seminal reviews and books related to this topic.
A further summary has been made, some duplicates and redundant elements have been removed, and some parts that are required for subsequent discussions in this article have been modified and highlighted in the original text. For example, the third paragraph of the fourth page, "RP gene deletions increase susceptibility to cancer", and the sixth page of "5.1" has been deleted in large quantities.
- Headers and footers suggest the manuscript is being considered in Cancers and not Biomolecules.
I'm sorry. Have been modified.
Yiwei Lu

Round 2
Reviewer 1 Report
I think the paper has been improved and that it is worthy being published in its present form.
Reviewer 2 Report
Accept in present form.